# Treatment of Primary Nummular Headache: A Series of 183 Patients from the NUMITOR Study

**DOI:** 10.3390/jcm12010122

**Published:** 2022-12-23

**Authors:** Cristina García-Iglesias, Francesca Puledda, Ana Echavarría-Íñiguez, Yesica González-Osorio, Álvaro Sierra-Mencía, Andrea Recio-García, Ana González-Celestino, Gonzalo Valle-Peñacoba, Pablo Irimia, Ángel Luis Guerrero-Peral, David García-Azorín

**Affiliations:** 1Headache Unit, Neurology Department, Hospital Clínico Universitario, 47003 Valladolid, Spain; 2Institute of Psychiatry, Psychology & Neuroscience, NIHR-Wellcome Trust King’s Clinical Research Facility/SLaM Biomedical Research Centre, King’s College London, London SE5 8AF, UK; 3Neurology Department, Clínica Universitaria de Navarra, Universidad de Navarra, 31008 Pamplona, Spain; 4Department of Medicine, Universidad de Valladolid, 47003 Valladolid, Spain

**Keywords:** headache disorders, nummular headache, epicranial headache

## Abstract

Nummular headache (NH) is a primary headache characterized by superficial coin-shaped pain. NUMITOR (NCT 05475769) is an observational study evaluating the responder rate of preventive drugs in NH patients. The treatment response was assessed between weeks 8 and 12 compared with the baseline. Patients were included between February 2002 and October 2022. Demographic and clinical variables were assessed; treatment response was estimated by 50%, 30%, and 75% responder rates and treatment discontinuation due to inadequate tolerability. A total of 183 out of 282 patients fulfilled eligibility criteria and completed the study. Patients were aged 49.5 (standard deviation (SD): 16.8) years, and 60.7% were female. NH phenotype was a parietal circular pain of four centimeters’ diameter, moderate intensity, and oppressive quality. At baseline, patients had 25 (interquartile range) pain days per month. Preventive treatment was used by 114 (62.3%) patients. The highest 50% and 75% responder rates corresponded to onabotulinumtoxinA (62.5%, 47.5%), followed by gabapentin (43.7%, 35.2%). Oral preventive drugs were not tolerated by 12.9–25%. The present study provides class IV evidence of the effectiveness of oral preventive drugs and onabotulinumtoxinA in the treatment of primary NH. OnabotulinumtoxinA was the most effective and best-tolerated drug, positioning it as first-line treatment of NH.

## 1. Introduction

In 2002, Pareja et al. reported the first series of 13 patients affected by a circumscribed coin-shaped cephalalgia that was named “nummular headache” (NH) [1]. In 2004, the disorder was included in the International Classification of Headache Disorders (ICHD) [2], being part of group 4—other primary headache disorders—in the third edition [3]. Although preventive treatment is required by 50–80% of patients with NH [4,5,6,7], evidence regarding this aspect is limited. As there are no existing randomized controlled trials, treatment recommendations are based on case reports, case series [5,6], or retrospective cohort studies [7], and propose gabapentin as first-line therapy and onabotulinumtoxinA for treatment-resistant patients. In addition, the definition of treatment response is heterogeneous, with studies accepting any degree of clinical benefit as positive response [4,5,6,7,8,9,10]. Further, no treatment recommendations regarding the design of studies evaluating the efficacy of preventive treatment of NH exist as they do for migraine [11]. In fact, for migraine trials, the three recommended efficacy endpoints are: (1) the reduction in the mean number of headache days per month, compared with the baseline; (2) the reduction in the mean number of migraine days per month, compared with the baseline, and (3) the 50% responder rate.

The main objective of the present study was to evaluate the 50% responder rate in patients with NH treated with preventive therapies between weeks 8 and 12 of treatment compared with baseline. The secondary endpoints included the evaluation of the 30% and 75% responder rates between weeks 8 and 12 and 20 and 24. The frequency and type of treatment-emergent adverse effects (TEAE) and treatment discontinuation due to adverse effects (AEs) were evaluated.

## 2. Materials and Methods

### 2.1. Study Design and Registry

The NUMITOR study (Nummular Headache Iberian Study on the Treatments and Outcomes in Real-World Setting) is an analytical observational study with an ambisective (retrospective and prospective) cohort design. Here, we report the data from the retrospective analysis of the study of prospectively collected patients from our registry. The study protocol and the statistical analysis plan (SAP) were published in ClinicalTrials.gov (NCT 05475769). The study was performed and reported in accordance with the Strengthening the Reporting of Observational Studies in Epidemiology (STROBE) guidelines [12].

### 2.2. Study Location and Participants

The study was carried out in the Hospital Clínico Universitario de Valladolid, a third-level public university hospital located in Valladolid. The East Valladolid Ethics Review Board approved the study (PI-GR-21-2394), and all participants read and signed an informed consent form.

### 2.3. Study Subjects and Eligibility Criteria

The study population were patients with NH who, under the opinion of their responsible neurologists, required preventive treatment in any of its modalities, oral or injectable.

The inclusion criteria were: (1) NH diagnosis according to the ICHD-3 criteria [2], (2) minimum duration of NH of 3 months or more, (3) age over 18 years old, and (4) informed consent signature.

The exclusion criteria were: (1) the diagnosis was better accounted for by any other disorder included in ICHD-3 [2]; (2) had any serious systemic or psychiatric pathology that makes it difficult to assess the patient; (3) had a secondary cause of NH [13], including post-traumatic NH; (4) used preventive drugs for another indication other than NH (e.g., epilepsy, other painful conditions, aesthetic, sleep disorders); and (5) had multifocal NH.

### 2.4. Study Period

The study period covered from February 2002, the date of the NH description [1], to October 2022.

### 2.5. Variables

Data were collected by headache experts using a standardized questionnaire that was adapted from the Valladolid NH registry, whose results have been previously published [6,13,14,15]. The studied variables included:(a)Demographic variables: sex, age of onset (years), age at the time of consultation, months of evolution from the NH onset to the NH diagnosis. Prior history of patients included prior history of hypertension, dyslipidemia, diabetes, overweight, smoking habit, alcohol abuse, asthma, nephrolithiasis, chronic painful syndromes, cardiovascular diseases, peptic ulcer disease, affective disorders, and sleep disorders. Prior history of other headache disorders was specifically assessed, including the type, as per ICHD-3 criteria [2].(b)Clinical variables: location (frontal, temporal, parietal, occipital, vertex), laterality (right, left, or parasagittal), shape (ovoid or circular), size (centimeters), baseline intensity (verbal analog scale (VAS) 0–10), quality of the pain (oppressive, throbbing, stabbing, electric, or others), worsening of pain with physical activity, presence of associated symptoms (photophobia, phonophobia, osmophobia, nausea, vomiting, cranial autonomic symptoms), presence of allodynia, presence of dysesthesia, presence of trophic changes, presence of remissions, and presence of exacerbations.(c)Paraclinical variables: presence of alterations in acute phase reactants (C-reactive protein (CRP), and erythrocyte sedimentation rate (ESR)) and presence of alterations in neuroimaging.(d)Treatment-related variables: symptomatic and preventive treatment use and response to preventive treatment, measured according to the criteria of the International Headache Society [11] by estimating the 30%, 50%, and 75% responder rates. The proportion of patients that discontinued each treatment due to lack of tolerability was estimated. Treatment was selected as per responsible physician criteria and the local standard of care and national guidelines [16]. OnabotulinumtoxinA (AbbVie Inc., North Chicago, IL, USA) was injected in five points in the painful area, with 5 units per point.

### 2.6. Information Sources and Data Collection

Electronic or paper medical records were reviewed. In case of missing data, participants were asked to contact patients and retrieve these by a clinical interview, whenever feasible. In those patients who were diagnosed before the publication date of ICHD-3 [3], the NH diagnosis was reviewed based on the criteria of the latest edition to increase consistency and guarantee comparability. A headache diary was given to participants to register the frequency of headache and the treatment need and response.

The information was completed in an anonymized, centralized database using a data collection form in the REDCap software (Yale University, New Haven, CT, USA).

### 2.7. Sample Size

Based on a preliminary analysis of data from our center [17], the 50% responder rate between weeks 8 and 12 of the main preventive treatments ranged between 50% and 77% for amitriptyline, lamotrigine, pregabalin, gabapentin, and onabotulinumtoxinA. With a 95% confidence level, for an estimated proportion of 66% and a precision of 10%, the sample size would be 84 patients, increasing to 98 with an expected proportion of patient losses of 15%.

### 2.8. Statistical Analysis

The SAP was published on 11 July 2022 at ClinicalTrials.gov. The ordinal qualitative and quantitative variables were presented as frequency and percentage, and quantitative variables as mean and standard deviation, or as median and interquartile range based on the type of distribution. The normality of the distribution was evaluated using the Kolmogorov–Smirnov test, and the homogeneity of variances by Levene’s test.

The statistical analysis was performed by intention to treat. To evaluate the responder rate, the percentage of patients who presented a reduction in the number of headache days per month of at least 30%, 50%, and 75% between weeks 8 and 12, compared with the month prior to the start of treatment, was calculated. The proportion of patients experiencing adverse effects and discontinuing treatment because of them was assessed. Missing data were addressed by conservative analyses by the baseline carried forward method. A statistical significance level of *p* < 0.05 was considered. For the adjustment for multiple comparisons, the false discovery rate according to the Benjamini–Hochberg method was used. Statistical Package for the Social Sciences (SPSS) (version 26.0 for Mac) (IBM Corp., Armonk, NY, USA) was used for the analysis.

## 3. Results

During the study period, *n* = 282 patients were screened. Of these, *n* = 269 fulfilled the inclusion criteria; a further *n* = 86 were excluded, with *n* = 183 fulfilling the eligibility criteria. Figure 1 shows the flow diagram of patients.

### 3.1. Demographic Variables and Prior Medical History

Regarding demographic variables, *n* = 111 (60.7%) patients were female, aged 49.5 (SD: 16.8) years at onset and 51.4 (SD: 16.9) years at consultation, with a median time of 7.5 (IQR: 3–14.2) months of evolution between NH onset and diagnosis. Concerning prior medical history, *n* = 164 patients had medical comorbidities (89.6%), including arterial hypertension in *n* = 62 (33.9%), dyslipidemia in *n* = 65 (35.5%), diabetes in *n* = 12 (6.6%), overweight in *n* = 48 (26.2%), smoking habit in *n* = 32 (17.4%), asthma in *n* = 2 (6.6%), chronic painful syndromes in *n* = 8 (4.4%), cardiovascular disorders in *n* = 55 (30.1%), peptic ulcer disease in *n* = 19 (10.4%), affective disorders in *n* = 49 (26.8%), and sleep disorders in *n* = 5 (2.7%). No patients had nephrolithiasis or alcohol abuse. Patients had prior history of headache disorders in *n* = 60 (32.8%) cases, including migraine in *n* = 28 (15.3%), epicrania fugax in *n* = 20 (10.9%), tension-type headache in 7 (3.8%), cranial autonomic cephalalgia in *n* = 2 (1.1%), primary headache associated with sexual activity in *n* = 1 (0.5%), primary stabbing headache in *n* = 1 (0.5%), and headache attributed to spontaneous intracranial hypotension in *n* = 1 (0.5%).

### 3.2. Clinical and Paraclinical Variables

Table 1 summarizes the clinical variables of the sample. The most frequent headache phenotype was parietal circular pain, with a median size of 4 centimeters, with moderate intensity, oppressive quality of pain, and rarely associated symptoms. Seven patients had increased ESR, and no patients had increased CRP upon diagnosis. Twenty-two patients had incidental findings in imaging studies that were deemed unrelated to NH by the responsible physician.

### 3.3. Treatment of NH

Patients required acute treatment in *n* = 141 (77.7%) cases and preventive treatment in *n* = 114 (62.3%) cases. The most frequently used preventives were gabapentin in *n* = 71 (38.8%) cases, onabotulinumtoxinA in *n* = 39 (21.3%), amitriptyline in *n* = 31 (16.9%), lamotrigine in *n* = 23 (12.6%), pregabalin in 9 (4.9%), anesthetic blockades in 9 (4.9%), beta-blockers in 8 (4.4%), flunarizine in 7 (3.8%), topiramate in 6 (3.3%), carbamazepine in 4 (2.2%), and mirtazapine in 1 (0.5%). Information about treatment response missed in cases of patients treated with amitriptyline (*n* = 2), onabotulinumtoxinA (*n* = 1), and gabapentin (*n* = 2), which were imputed by baseline carried forward. Table 2 lists the response to preventive drugs.

The drug with the highest 30%, 50%, and 75% response rate was onabotulinumtoxinA. Figure 2 shows the 30%, 50%, and 75% responder rate for the most frequently used drugs.

## 4. Discussion

The present study represents the largest published series of real-world evidence regarding preventive treatment of primary NH. IHS-recommended endpoints of efficacy were employed in order to ensure the highest possible quality of evidence and in order to allow the comparability and combination of our results in future studies and meta-analyses. The main results of our study were in line with the existing literature on NH, with onabotulinumtoxinA and gabapentin found to be the most effective treatments. With regard to tolerability, oral preventive drugs were not tolerated by 12–25% of patients, leading to treatment discontinuation. In this regard, onabotulinumtoxinA was also better tolerated.

The demographic profile of patients with NH differs from that of those with migraine, with an older age and higher frequency of comorbidities [18]. In our study, we specifically assessed the frequency of specific comorbidities that would contraindicate the use of oral preventive drugs, such as asthma, nephrolithiasis, or cardiovascular disorders.

In the literature, five studies have described the benefit of onabotulinumtoxinA in NH, finding it effective in 42/64 of treated patients [10,19,20,21,22]. This drug was first used in the treatment of NH in 2008 in a series of 4 patients who showed benefit with the treatment [19]. A series of 5 NH patients who were resistant to other oral preventive drugs and anesthetic blockades was published subsequently; of these, 3 improved with onabotulinumtoxinA treatment [20]. Further case reports [21,22] and case series expanded the topic, with the largest series (*n* = 53) being published by our group, where a 50% responder rate of 62% between weeks 8 and 12 of treatment, compared with the baseline, was observed in NH patients [10]. In this study, we were able to compare the efficacy and tolerability of onabotulinumtoxinA with other oral preventive drugs for the first time with onabotulinumtoxinA showing better results. This is particularly relevant in a population that is on average older than that of other primary headache disorders, and with more frequent comorbidities. For these reasons, we suggest guidelines proposing onabotulinumtoxinA as the first drug of choice in the treatment of NH [16].

Gabapentin is the oral preventive drug with most available evidence in the treatment of primary NH. Case reports and case series have shown pain disappearance in 27/43 (40%) [5,23,24,25,26,27,28] of NH patients treated with gabapentin. Improvement in the intensity of pain was found in 18/39 (46%) patients [5,25,29,30], while no response was reported in 5/6 (83%) patients [21,30,31] and lack of tolerability in 1 additional case [32]. These results could indeed be due to a publication bias, with positive studies more likely to report the clinical benefit; further, only two large series have been published in the literature so far [5,6]. In our study, 21% of NH patients treated with gabapentin discontinued the drug due to inadequate tolerability. The proportion of patients with migraine who discontinued gabapentin due to tolerability issues was even higher, at 25% (24/98) [33], with a higher proportion of adverse events observed in patients treated with higher gabapentin doses of up to 1200–3000 mg per day [34].

Evidence supporting the use of amitriptyline is even more limited, with only 4/6 (67%) patients describing pain disappearance [7]. Pain benefit in previous reports was shown in 2/6 (33%) patients [7], partial benefit after a combination of drugs including amitriptyline in three additional patients [5], and no response in 6/15 (40%) patients [5,35]. There was no evidence supporting the use of lamotrigine in NH [6], this report being the first series that show positive benefit in some cases. This drug could indeed represent an alternative in patients with resistance or contraindication to other preventive drugs. The effect of other therapies, such as magnesium or vitamin D, has not been explored yet in NH, but could potentially have a role [36].

The main limitations of this study lie within its retrospective nature. There was also the possibility of selection bias, as we tended to include patients with more severe NH, while patients with mild or inactive NH were less likely to be represented. To minimize detection bias and increase the validity of the study, a standardized questionnaire was used to minimize interobserver variability. To avoid attrition bias and minimize the impact of missing data, conservative analyses and assumptions were made. Future studies should expand the treatment response beyond 12 weeks of treatment. The NUMITOR study will provide further evidence from multiple centers with a prospective cohort design.

## 5. Conclusions

The present study provides class IV evidence on the treatment of primary nummular headache. The 50% responder rate between weeks 8 and 12, compared with the baseline, ranged from 35–44% following oral preventive drugs to 62% after onabotulinumtoxinA use. OnabotulinumtoxinA also showed the highest 75% responder rate and tolerability rates, with no patients discontinuing treatment because of side effects, as opposed to oral preventive treatments, which were discontinued due to poor tolerability in 12–25% of patients. These findings support the use of onabotulinumtoxinA as first-line treatment of NH.

The study was funded by the Regional Health Administration (Gerencia Regional de Salud, SACYL, Castilla y León, GRS 2416/A/21).

## Figures and Tables

**Figure 1 jcm-12-00122-f001:**
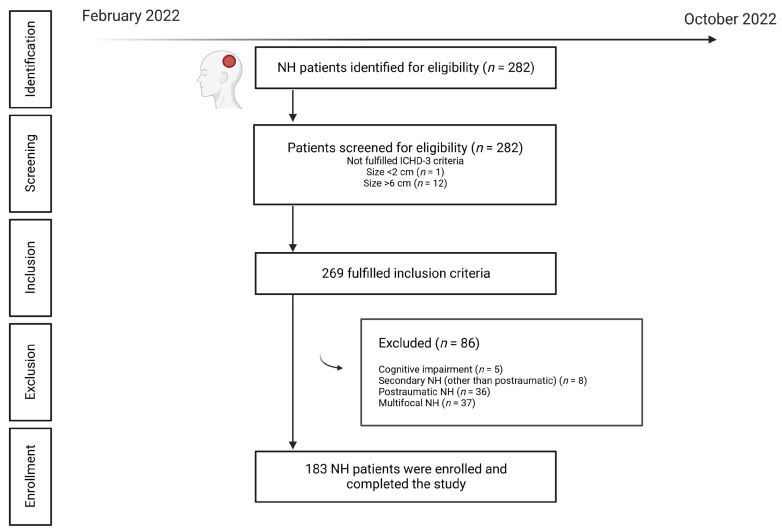
Flow diagram of screened, included, excluded, and enrolled patients.

**Figure 2 jcm-12-00122-f002:**
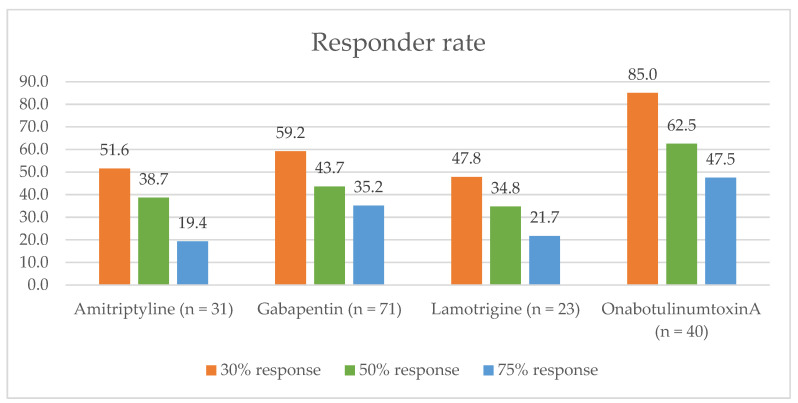
Responder rates of 30%, 50%, and 75%.

**Table 1 jcm-12-00122-t001:** Clinical variables of patients with NH.

Variable	Frequency (%)
Location *	
Frontal location	45 (24.6%)
Temporal location	27 (14.8%)
Parietal location	77 (42.1%)
Occipital location	41 (22.4%)
Vertex location	7 (3.8%)
Facial location	0 (0%)
Laterality	
Right sided	86 (50.4%)
Left sided	70 (38.3%)
Parasagittal	27 (14.8%)
Ovoid shape	26 (14.2%)
Circular shape	157 (85.8%)
Size (cm)	4 (IQR: 3–5)
Baseline intensity (0–10)	5 (IQR: 4–6)
Headache days per month (days)	25 (IQR: 10–30)
Quality of pain	
Oppressive quality	78/154 (50.6%)
Throbbing quality	13/154 (8.4%)
Stabbing quality	30/154 (19.5%)
Electric quality	0 (0%)
Burning quality	33/154 (21.4%)
Worsening by physical activity	10 (5.5%)
Associated photophobia	10 (5.5%)
Associated phonophobia	8 (4.4%)
Associated osmophobia	0 (0%)
Associated nausea	8 (4.4%)
Associated vomiting	6 (3.3%)
Cranial autonomic symptoms	4 (2.2%)
Presence of allodynia	65 (35.5%)
Presence of dysesthesia	59 (32.2%)
Presence of trophic changes	3 (1.6%)
Presence of remissions	20 (10.9%)
Presence of exacerbations	88 (48.1%)

* When NH was located in between two bones, both locations were selected (i.e., parieto-occipital). Differing denominator indicates missing data.

**Table 2 jcm-12-00122-t002:** Treatment response to preventive drugs.

Drug	Not Tolerated	No Response (0–30%)	Partial Response (31–50%)	Adequate Response (51–75%)	Optimal Response (>75%)
Amitriptyline (*n* = 31)	4 (12.9%)	11 (35.5%)	4 (12.9%)	6 (19.3%)	6 (19.3%)
Beta-blockers (*n* = 8)	1 (12.5%)	4 (50%)	1 (12.5%)	2 (25%)	0 (0%)
Anesthetic blockades (*n* = 9)	0 (0%)	1 (11.1%)	2 (22.2%)	1 (11.1%)	5 (55.6%)
Carbamazepine (*n* = 4)	1 (25%)	1 (25%)	0 (0%)	0 (0%)	2 (50%)
Flunarizine (*n* = 7)	0 (0%)	5 (71.4%)	2 (15.5%)	0 (0%)	0 (0%)
Gabapentin (*n* = 71)	15 (21.1%)	14 (19.7%)	11 (15.5%)	6 (8.5%)	25 (35.2%)
Lamotrigine (*n* = 23)	3 (13.0%)	9 (39.1%)	3 (13.0%)	3 (13.0%)	5 (21.7%)
Mirtazapine (*n* = 1)	0 (0%)	1 (100%)	0 (0%)	0 (0%)	0 (0%)
Pregabalin (*n* = 9)	2 (22.2%)	1 (11.1%)	4 (44.4%)	1 (11.1%)	1 (11.1%)
Topiramate (*n* = 6)	0 (0%)	5 (83.3%)	0 (0%)	1 (16.7%)	0 (0%)
OnabotulinumtoxinA (*n* = 40)	0 (0%)	6 (15.0%)	9 (22.5%)	6 (15.0%)	19 (47.5%)

## Data Availability

The data are available upon request from the corresponding author.

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
