# Peer review of "Treatment of Primary Nummular Headache: A Series of 183 Patients from the NUMITOR Study"

_jcm, 2022, doi:10.3390/jcm12010122_

Round 1

Reviewer 1 Report

The manuscript is interesting; the text is clear and easy to follow. Also the data are really clear. I have only few poiints for the aithors:

- Introduction: I suggest to expand it in order to explain the recent advances about headache: please, read and cite the paper by Dell Isola GB et al. J Clin Med 2021 Dec 20; 10(24):5983

Methods: I suggest to define better the inclusion criteria and to report in detail all points (as done for the exclusion criteria).

Reviewer 2 Report

In a retrospective study, authors could evaluate the effectiveness of available treatments for nummular headache. It is an interesting study. There are however some issues that need to be addressed before this manuscript can be considered for publication (see my Comments below).

Abstract

-I suggest to add more details of methods to the Abstract.

- According to the obtained results, the appropriate treatment of this group of patients can be suggested in conclusion.

Main text

-The available and first drug of choice for the treatment of NH patients in the clinic can be added.

-Authors mentioned the adverse effects of drugs were evaluated. I could not find any result to support this part.

-The appropriate treatment of NH patients can be suggested in conclusion with more details.

Round 2

Reviewer 2 Report

The authors have addressed all my comments for this paper.